# Inflammatory Immune Responses in the Pathogenesis of Tick-Borne Encephalitis

**DOI:** 10.3390/jcm8050731

**Published:** 2019-05-22

**Authors:** Petra Bogovič, Lara Lusa, Miša Korva, Miša Pavletič, Katarina Resman Rus, Stanka Lotrič-Furlan, Tatjana Avšič-Županc, Klemen Strle, Franc Strle

**Affiliations:** 1Department of Infectious Diseases, University Medical Center Ljubljana, Japljeva 2, 1525 Ljubljana, Slovenia; stanka.lotric-furlan@mf.uni-lj.si (S.L.-F.); franc.strle@kclj.si (F.S.); 2Institute for Biostatistics and Medical Informatics, Faculty of Medicine, University of Ljubljana, Vrazov trg 2, 1000 Ljubljana, Slovenia; lara.lusa@mf.uni-lj.si; 3Department of Mathematics, Faculty of Mathematics, Natural Sciences and Information Technologies, University of Primorska, Glagoljaška 8, 6000 Koper, Slovenia; 4Institute for Microbiology and Immunology, Faculty of Medicine, University of Ljubljana, Zaloška 4, 1000 Ljubljana, Slovenia; Misa.Korva@mf.uni-lj.si (M.K.); Misa.Pavletic@mf.uni-lj.si (M.P.); katarina.resman@mf.uni-lj.si (K.R.R.); Tatjana.Avsic@mf.uni-lj.si (T.A.-Z.); 5Division of Rheumatology, Allergy, and Immunology, Center for Immunology and Inflammatory Diseases, Masachusetts General Hospital/Harvard Medical School, 55 Fruit Street, Boston, MA 02114, USA; KSTRLE@mgh.harvard.edu

**Keywords:** tick-borne encephalitis, inflammatory mediators, innate immunity, adaptive immunity, cytokines, chemokines, cerebrospinal fluid, severity of illness

## Abstract

Clinical manifestations of tick-borne encephalitis (TBE) are thought to result from the host immune responses to infection, but knowledge of such responses is incomplete. We performed a detailed clinical evaluation and characterization of innate and adaptive inflammatory immune responses in matched serum and cerebrospinal fluid (CSF) samples from 81 adult patients with TBE. Immune responses were then correlated with laboratory and clinical findings. The inflammatory immune responses were generally site-specific. Cytokines and chemokines associated with innate and Th1 adaptive immune responses were significantly higher in CSF, while mediators associated with Th17 and B-cell responses were generally higher in serum. Furthermore, mediators associated with innate and Th1 adaptive immune responses were positively associated with disease severity, whereas Th17 and B cell immune responses were not. During the meningoencephalitic phase of TBE, innate and Th1 adaptive inflammatory mediators were highly concentrated in CSF, the site of the disease. The consequence of this robust immune response was more severe acute illness. In contrast, inflammatory mediators associated with B cell and particularly Th17 responses were concentrated in serum. These findings provide new insights into the immunopathogenesis of TBE and implicate innate and Th1 adaptive responses in severity and clinical presentation of acute illness.

## 1. Introduction

Tick-borne encephalitis (TBE), an infection of the central nervous system (CNS), is endemic in many regions in Europe and Asia. It is caused by three subtypes of the TBE virus (TBEV)—European, Siberian, and Far-Eastern—and is transmitted to humans predominantly through *Ixodes* spp. tick bites, and only rarely by consumption of infected milk or milk products (usually goat). Although no specific treatment for TBE is available, the disease is largely preventable by vaccination [1].

Similar to the other flaviviruses, only a small proportion (2–25%) of patients infected with TBEV develops symptoms. However, the illness can be quite serious. Most patients with symptoms caused by infection with the European TBEV subtype have a biphasic course of illness. The initial phase, which corresponds to viremia, presents as a febrile illness accompanied by fatigue, malaise, headache, and muscle and joint pain that last for up to 8 days, followed by improvement for about 1 week. The clinical hallmark of the second phase of TBE is CNS involvement. In children, meningitis is the predominant clinical manifestation of TBE. In adults, ~50% of patients develop meningitis, ~40% develop meningoencephalitis, and ~5–10% develop meningoencephalomyelitis. The case fatality rate of TBE caused by European subtype of TBEV is between 0.5 and 2%. In addition, ~5% of patients are affected by permanent pareses and at least 30% suffer from a postencephalitic syndrome [1,2,3,4].

Although the clinical presentation of acute illness has been well described, knowledge of pathogenesis, particularly as it relates to the range in severity of clinical signs/symptoms and the outcome of TBE, remains unclear. Moreover, the clinical manifestations of TBE are thought to be due at least in part to host immune response to the virus. However, the understanding of the immune responses and their impact on clinical presentation and outcome of TBE is limited.

In the present study, we performed a detail characterization of inflammatory immune responses in 81 well-defined adult patients with TBE in whom extensive clinical information was available. All patients were from Central Europe (Slovenia) and were thought to be infected with the European TBEV subtype. The levels of 24 cytokines and chemokines associated with innate and adaptive T and B cell immune responses were assessed in matched serum and cerebrospinal fluid (CSF) samples obtained at the beginning of the meningoencephalitic phase of TBE. The inflammatory immune profiles were correlated with the disease severity to gain insights into pathogenicity of TBE.

## 2. Materials and Methods

The study was approved by the Medical Ethics Committee of the Ministry of Health of the Republic of Slovenia (No 0120-467/2017/3, approved on 3 December 2017). Each participant provided written informed consent. The subjects’ consent was obtained according to the Declaration of Helsinki.

### 2.1. Evaluation of Patients 

This study is based on 420 adult patients who were diagnosed with TBE and assessed for long-term outcome at the Department of Infectious Diseases, UMC Ljubljana, Slovenia. Detailed clinical information and evaluation criteria on these patients were reported previously [3]. Briefly, TBE was defined as a febrile illness with clinical symptoms and/or signs of meningitis or meningoencephalitis, CSF pleocytosis (>5 × 10^6^ leukocytes/L), and demonstration of acute TBEV infection (presence of serum IgM and IgG antibodies to TBEV). Antibodies to TBEV were assessed using the Enzygnost^®^ Anti-TBE Virus (IgM, IgG) test (SiemensGmbH, Marburg, Germany) according to the manufacturer’s protocol. Demographic, epidemiologic, laboratory, and clinical data were obtained for all patients, thus enabling detailed appraisal of the course and severity of the acute illness, including daily evaluation of the presence and intensity of TBE signs and symptoms during the hospital stay, together with several blood and CSF laboratory analyses. Symptoms that newly developed or worsened since the onset of TBE, and which had no other known medical explanation, were interpreted as TBE-associated symptoms. On physical examination, particular attention was paid to signs of neurological involvement (tremor, ataxia, cranial and spinal nerve paralysis, etc.). 

The severity of acute illness was evaluated quantitatively using a standardized questionnaire, as reported previously [5]. The presence, intensity, and duration of an individual TBE-associated symptom or sign were scored on a scale of 1–9, with the absence of a particular symptom or sign assigned a score of zero; the severity score was defined as the sum of the individual scores. This approach allows for stratification of patients according to disease severity; corresponding to mild (score 0–8), moderate (score 9–22), or severe (score > 22) disease [5].

### 2.2. Selection of Patients for the Analysis of Cytokine and Chemokine Levels in Serum and CSF 

A total of 81 patients were selected for the study of immune responses, from the larger cohort of 420 patients evaluated in a previous study of long-term outcomes of TBE [3]. Patients were selected to represent a range of disease manifestations from mild to severe acute illness, as well as on the availability of matched CSF specimens and serum samples. Demographic and clinical characteristics in 81 patients who were selected for the present study were similar to the rest of the larger cohort.

For these studies, matched serum and CSF samples were obtained at the time of acute illness, during meningoencephalitic phase of TBE (Figure 1). Serum and CSF non-centrifugated samples from the same patients were obtained and processed on the same day within a time span of a few (as a rule < 3) h to minimize variation in comparing the immune responses between the two sites. These samples were aliquoted and frozen at −80 °C for an average of 79 (59–82) months prior to testing. To maintain integrity, samples did not undergo freeze-thaw prior to cytokine analysis.

### 2.3. Cytokine and Chemokine Determinations 

The levels of 24 cytokines/chemokines associated with innate (GMCSF, IFNα, IL-10, IL-1β, IL-6, IL-8, TNFα, CCL2, CCL3) and adaptive Th1 (IFNγ, IL-12P40, IL-12P70, CXCL10, CXCL9, CCL19), Th17 (IL-17F, IL-17A, IL-22, IL-21, IL-23, IL-25, IL-27), and B cell immune response (CXCL12, CXCL13) were assessed in matched patient CSF and serum samples using bead-based multiplex assays (Luminex, EMD Millipore, Burlington, Massachusetts, United States) on the MagPix system. To minimize inter-assay variation, all measurements in a single panel were performed on the same day in one complete experiment, according to manufacturer’s instructions, with final sample dilution of 1:5. For all plates in a single panel, simultaneous analysis was done with the Milliplex Analyst 5.1 software. Values outside the upper or lower end of the standard curve (ie, out-of-range values) were considered as maximum or minimum values, respectively.

### 2.4. Statistical Analyses 

Numerical variables were summarized with medians (interquartile ranges, IQR) and the categorical variables with frequencies and percentages (with their 95% confidence intervals (CIs)). The number of missing values was reported. 

The concentrations of the 24 cytokines/chemokines in paired CSF and serum were compared using Wilcoxon signed rank test. To control for false positives, the *p* values were adjusted using a multivariate permutation procedure [6].

The association between cytokine/chemokine levels and leukocyte counts in CSF or serum was assessed with Spearman’s rho rank-based correlation and the association was tested using Spearman’s method and adjusted for multiple comparisons [6]. The same approach was used to evaluate the association of cytokines/chemokines with the severity score, with the levels of IgG antibodies to TBE virus and with duration of meningoencephalitic phase of the illness. 

We displayed observed associations with outcome variables graphically by using box and whisker plots for categorical variables and scatter plots for numerical variables. We added a loess regression (locally weighted scatterplot smoothing) line with 95% CIs fitted by using the geom_smooth function in the ggplot2 R software.

Adjusted *p* values <0.05 were considered significant. Statistical analysis was performed using R statistical language (R), version 3.5.0. [7].

## 3. Results

Basic demographic, clinical, and laboratory information on the acute illness in 81 patients is shown in Table 1. Patients’ median age was 56 (IQR 43–63) years, 49.4% were males, and 39 (48.1%) had at least one underlying illness. There were no differences in underlying diseases between females and males. All the patients were tested for borrelial infection. Concomitant Lyme neuroborreliosis was established in three patients. The predominant feature of the acute illness was meningoencephalitis/meningoencephalomyelitis (59.6%). Duration of neurologic symptoms before CSF and blood sample obtained was 5 (3–6) days. Demographic and clinical characteristics in these 81 patients were similar to the larger cohort of 420 TBE patients reported previously [3].

### 3.1. Comparison of Cytokine and Chemokine Levels in Serum and CSF 

To characterize the inflammatory immune responses in patients with TBE, the levels of 24 cytokines and chemokines associated with innate or adaptive immune responses were assessed in matched serum and CSF samples obtained on the same day within a time span of 1–3 h. As shown in Table 2 and Figure 2 the levels of most (18 of 24) cytokines and chemokines tested differed substantially according to the compartment (CSF vs serum) from which the specimens were acquired, implying the localization of these responses at different sites. Moreover, this analysis revealed clear distinctions in the type of immune response present at each site. Cytokines and chemokines associated with innate immune responses or Th1 adaptive immune responses were markedly higher in CSF (except for TNFα). In contrast, mediators associated with Th17 and B cell immune responses were generally higher in serum. These findings imply that innate and Th1 adaptive responses are occurring locally in CNS, whereas Th17 and B cell responses may be triggered in systemic circulation.

The gray points represent the individual data values measured in CSF and serum. The superimposed boxplots represent the first quartile (i.e., lower edge of the box), median (i.e., bar inside the box), third quartile (i.e., upper edge of the box), and minimum and maximum (i.e., length of the whiskers). If any points are at a greater distance from the quartiles than 1.5 times the interquartile range (IQR), the whisker length represents a distance of 1.5 times IQR from the upper or lower quartile. 

### 3.2. Cytokine and Chemokine Levels According to White Cell Counts in Blood or CSF

Because of the differences in the localization of the immune responses, we then correlated the levels of cytokines and chemokines in blood or CSF with basic white cell counts at those sites. This allowed for direct comparison of immune mediators and corresponding white cell counts from the same patients, at the same site, and on the same day. 

Nearly all inflammatory mediators associated with the innate and Th1 adaptive responses correlated positively with CSF white cell concentrations (as indicated with positive rank correlations rho). In contrast, these associations were much less prominent and more diverse for cytokines and chemokines associated with Th17 or B cell immune responses. Because of the number of variables in each comparison we then performed statistical adjustment for multiple comparisons. Most of the associations remained statistically significant after adjustment for multiple comparisons: positive correlations between CSF levels of IL-1β, IL-6, and IL-8 and CSF neutrophils; TNF, IFNγ, CXCL9, CXCL10, CCL19, and IL-27 and lymphocyte concentrations; and CXCL10 and monocytes. In contrast, there was a negative correlation between the CSF levels of B cell chemoattractant, CXCL12, and CSF lymphocyte concentration, presumably because these responses tended to be concentrated in serum. Thus, in CSF, several inflammatory mediators, particularly those associated with innate and Th1 adaptive immune responses, correlated positively with leukocyte counts in CSF, demonstrating the localization of these cellular and corresponding inflammatory immune responses in the central site of disease in patients with TBE (Table 3, Appendix A).

In contrast to CSF, no significant associations were observed between the levels of cytokines/chemokines in serum and blood leukocyte counts, which could reflect the spatial and/or temporal discontinuum between immune cell frequency and accumulation of inflammatory mediators in blood (Table 3, Appendix A).

### 3.3. Comparison of Cytokine and Chemokine Levels in CSF and Serum According to Duration of Illness

Although samples were only available from the first visit, there was a range in the duration of meningoencephalitis at the time these samples were obtained (median 5, range 2–14 days, IQR 3–6 days) due to different duration of CNS involvement before admission to the hospital. This range provided an opportunity to gain insight into dynamic interplay between inflammatory immune response and the duration of the early meningoencephalitic phase of the illness. 

In general, CSF cytokines and chemokines reached the highest levels on days 5–7 of the meningoencephalitic phase and decreased thereafter. However, the association of cytokine/chemokine levels with the duration of illness was statistically significant only for B cell chemoattractant, CXCL12 (*p* = 0.003) and a Th17-associated cytokine, IL-27 (*p* = 0.025). Dynamics of the CSF chemokines/cytokines is depicted in Figure 3. No corresponding associations were found in serum (Appendix A).

The black dots represent individual values of the duration of early meningoencephalitic phase of the illness and the cytokine and chemokine levels in CSF for each patient. The (loess) blue curves estimate the functional association between the two variables using a non-parametric smoother, the gray bands are their 95% CIs.

### 3.4. Association of Inflammatory Mediators with Severity of Acute Illness

Although the host immune response has been implicated in disease severity in TBE, systematic studies of inflammatory mediators in relation to disease severity are limited [8]. Herein we assessed a range of innate and adaptive inflammatory mediators according to disease severity which was defined using severity scoring based on a standardized questionnaire, as reported previously [5].

As indicated by positive rho rank correlation values, most mediators associated with the innate and Th1 immune response were positively associated with disease severity. These associations were in general more prominent in CSF than in serum. Many of the associations remained statistically significant after adjustment for multiple comparisons including: positive correlations between greater disease severity score and higher CSF levels of IL-1β (*p* = 0.02), CCL3 (*p* = 0.03), IL-12P40 (*p* = 0.01), and IL-12P70 (*p* = 0.03); and serum levels of CXCL9 (*p* = 0.002) and CXCL10 (*p* = 0.04). The associations between disease severity and Th17 and B cell immune responses were modest and generally negative (Table 4). These correlations suggest that both innate and Th1 adaptive immune responses may play a role in the pathogenesis of clinical disease during acute infection.

### 3.5. Association of Inflammatory Mediators with Levels of IgG Antibodies against TBEV in CSF and Serum

As shown in Table 5, rho rank correlation values indicated that the majority (6/9) of mediators representing the innate immune response in CSF were negatively associated with the levels of IgG antibodies against TBEV in CSF. In contrast, the association between TBEV antibodies and cytokines/chemokines representing adaptive immune responses were predominantly positive (Th1: 4/6, Th17: 2/3, B cell: 2/2); for two of them (IL-27 and CXCL12) the associations remained statistically significant after adjustment for multiple comparison. In contrast, the corresponding associations in serum were negative; however, only association with CCL3 remained statistically significant after adjustment for multiple comparisons (Table 5). The associations for individual patients are shown in Appendix A.

## 4. Discussion

Host immune response is thought important for controlling TBE infection and for the resulting clinical manifestations of this disease, but the understanding of TBE pathogenesis remains incomplete [8,9]. Emerging evidence in animal models and in patients demonstrates that both cellular and humoral immune responses are triggered to TBEV infection. As with other flaviviruses, studies in mice suggest that a robust humoral immune response is critically important in controlling TBEV infection [10,11,12,13]. The E protein of the TBEV, which is important for the attachment and entry of the virus into cells and for fusion with endosomal membranes, is the major target and inducer of neutralizing antibodies [14]. Studies of neutralizing antibodies in experimentally infected laboratory animals [15,16,17] and in TBE patients [18] suggest that individuals with mild TBE have higher concentrations of antibodies than those with a severe form of the disease, implying their role in disease pathogenicity. This distinction was clearly evident also in a recent study of adult patients with TBE, in which a significant negative association was observed between the level of specific TBEV serum IgG antibodies (during the initial examination of the meningitic/meningoencephalitic phase) and the severity of acute illness [19]. 

In addition to humoral responses, effective innate and adaptive cellular immunity is required for clearance of established infection. As is typical of viral infections, robust innate immune responses dominated by type I IFN signature appear to be an important first line of defense in TBEV infection. In cell cultures experiments using a human neuronal cell line, infection with TBEV induced a wide range of interferon-stimulated genes and pro-inflammatory cytokines, and pre-treatment with type I IFN inhibited TBEV production [20]. In addition to type I IFN signature, dendritic cells or astrocytes infected with TBEV also produce higher levels of IL-1β, TNF, IL-6, and IL-8, prototypical innate inflammatory mediators, and CXCL10, a potent chemoattractant for CD4+ T effector cells, particularly Th1 [21]. Similarly, studies of inflammatory mediators in patients with TBE demonstrate that TBEV infection is associated with a broad spectrum of cytokines and chemokines reflective of innate (IL-6, IL-8), T cell adaptive Th1 (IFNγ) and Th17 (IL-17F) responses, and B cell (CXCL13) responses. Collectively, these studies demonstrate that a wide range of innate and adaptive T cell (Th1 and Th17) and B cell immune responses are triggered in response to TBEV and likely play a role in the control of TBEV infection. 

This robust host immune response to TBEV infection appears to shape the clinical presentation of disease. Although comprehensive studies of the role of host immune responses in the clinical course and outcome of TBE are lacking, a few studies have now implicated host immune responses in disease presentation and immunopathology [22,23,24]. In a study of inflammatory responses and viral antigen loads in post-mortem brains of patients with fatal TBE, an inverse topographical correlation was observed between severe inflammatory changes in the tissue and the distribution of viral antigen [25]. Moreover, cytotoxic T cells were observed in contact with infected neurons. These findings suggested that although the host immune response is important in controlling TBEV infection, the consequence of this robust immune response is a more difficult clinical course of disease, presumably due to inflammation-induced tissue pathology. Nevertheless, the knowledge of broad-range of inflammatory immune profiles in patients with TBE is limited, and it is not yet well-understood if and how such responses influence the disease course and outcome.

Our current study builds upon these concepts by comprehensive characterization of the inflammatory immune responses in TBE patients with a range of disease severity to better understand the role of the immune responses in the pathogenesis of TBE. For this purpose, we assessed the levels of 24 inflammatory mediators associated with innate and adaptive T and B cell responses in matched serum and CSF samples in a cohort of 81 patients with TBE in whom detailed clinical information was available. The inflammatory responses in serum and CSF were correlated with immune cell findings and with clinical information, to identify the type of immune responses associated with the severity of illness. To help interpret the results, we grouped the cytokines and chemokines according to the function they are most commonly associated with. Although most inflammatory mediators may have overlapping functions, we think there is value in classifying the mediators according to their main biological function. In addition, we think that it adds validity to the findings if several mediators of a particular immune response, rather than only one, were associated with a particular outcome, such as the localization of Th1 mediators within CSF, the site of disease, and their association with disease severity.

The design of this study enabled an in-depth examination of innate and adaptive (T and B cell) inflammatory immune responses early in the course of central nervous system involvement, during the initial meningoencephalitic phase of TBE (median 5 days, range 2–14) when patients are usually admitted to hospital (Figure 1). Comparison of serum and CSF revealed two distinct types of immune responses in the two compartments. Cytokines and chemokines associated with innate and Th1 adaptive immune responses were concentrated in CSF, while mediators associated with Th17 and B cell immunity were generally higher in serum (Table 2, Figure 2). These findings implied that during the early stage of CNS involvement, innate and Th1 adaptive immune responses are occurring locally within CSF. In contrast, Th17 and B cell responses appear to be occurring within systemic circulation. However, because specimens from the initial phase of TBE prior to TBEV infection of the CNS were not available, we cannot exclude the possibility that the differences between findings in serum and CSF reflect at least in part the temporal lag between the events in the two different compartments; e.g., innate and Th1 responses found in CSF had developed initially in blood prior to the meningoencephalitic phase of TBE, and then localized to CSF as the infection crossed the blood–brain barrier. Studies of patients during the early phase of TBE would be needed to determine the initial immune events in this infection, however it is difficult to capture such patients since most does not seek medical care or are not recognised as having the initial phase of TBE and are thus not diagnosed with TBEV infection until the meningoencephalitic phase of the disease. 

Studies of cellular immune responses and inflammatory responses in human patients during the initial phase of TBE are lacking. However, several articles have characterized the inflammatory responses to TBEV in the second, meningoencephalitic phase of the disease [26,27,28,29,30,31,32]. Three recent studies assessed the levels of innate or Th17 adaptive inflammatory mediators in serum or CSF in 15 patients with TBE and compared them to six patients with non-TBE meningitis or eight patients with non-meningitis controls [29,33,34]. Collectively, these studies showed that the levels of innate immune mediators, particularly neutrophil chemoattractants (IL-8, CXCL1, CXCL2) and intrathecal expression of cytokines associated with Th17 responses (interleukin-17A, IL-17F, IL-22), in patients with TBE were higher compared to control groups [33]. Moreover, the authors concluded that Th17 cytokine responses are present in TBE and may contribute to the CNS neutrophilic inflammation [34]. Another group [24] performed a comprehensive analysis of the immune responses in serum of 87 patients with TBE and also showed that of the 30 cytokines, chemokines, and growth factors assessed, the levels of proinflammatory cytokines IL-6, IL-8, and IL-12 were significantly higher in patients than in control subjects. Finally, Günther and coworkers investigated several cytokines associated innate and Th1 adaptive immune responses (IL-10, IFNγ, IL-1ra, TNFα, and IL-6) in CSF and serum in relation to the aetiology of meningitis and clinical course in 44 patients with TBE and 36 patients with aseptic meningoencephalitis of another aetiology (non-TBE group). Similarly to our findings, CSF levels of TNFα were lower than in serum, whereas CSF levels of IFNγ, IL-10, and IL-6 exceeded the corresponding serum levels in the majority of patients with TBE [35]. However, because most studies to date consisted of small patient groups often lacking detailed clinical information and evaluated only a limited number of inflammatory mediators in serum or in CSF, but rarely in both sites, knowledge of the role of immune responses in the clinical course and outcome of TBE is incomplete.

Despite these limitations, the reported findings are largely consistent with the findings in our study in which both innate and adaptive T cell (Th1, Th17) and B cell inflammatory immune responses were detected in patients with TBE. A comparison of the levels of these mediators in matched CSF and serum samples demonstrated significantly greater levels of innate and adaptive Th1 mediators in CSF, including a potent neutrophil chemoattractant IL-8, and prototypical Th1 effector cytokine IFNγ and IFNγ-inducible chemoattractants CXCL9 and CXCL10. This suggests that during the meningoencephalitic phase of illness innate and Th1 adaptive immune responses are concentrated in the CNS, presumably to fight the TBEV infection which has crossed blood–brain barrier into CNS. In contrast, B cell chemoattractants CXCL12 and CXCL13, and most (6 of 7) Th17 associated cytokines were found in equal or higher concentrations in serum than in CSF, suggesting that these responses may be occurring outside the blood–brain barrier, presumably in systemic circulation or for Th17 responses in the mucosa. It is of interest that somewhat analogous findings were observed in studies of immune responses in serum and joint fluid samples in patients with Lyme arthritis, which is caused due to infection with tick-borne spirochete, *Borrelia burgdorferi*. In Lyme arthritis, Th1 responses were highly concentrated in the joint, the site of disease, whereas Th17 mediators were generally similar or even higher in serum. Moreover, the Th17 responses correlated directly with antibody responses implying a close link between Th17 immunity and specific B cell responses [36,37], whereas Th1 associations with antibody responses were more limited. Collectively, this suggests that Th1 responses have a more direct effector function in TBE, whereas Th17 responses are generated elsewhere and likely serve a different function; perhaps by promoting B cell responses.

Emerging evidence has implicated elevated inflammatory responses in greater severity of acute disease. Günther and coworkers reported that TBE patients with encephalitis had significantly lower levels of IL-10, a potent anti-inflammatory cytokine, in CSF (days 7–18 of the clinical course of illness) than TBE patients with meningeal disease, but similar levels of IFNγ, IFNα, and IL-6, implying that the lower levels of IL-10 are not able to sufficiently control inflammatory responses in CSF, thereby contributing to pathophysiology of TBE [35]. However, a more recent study of inflammatory responses in TBE patients showed no association between CSF or serum levels of MIF or TNFα with the severity of clinical signs and symptoms of TBE, including the presence or lack of mental status alterations, and paresis [33]. However, in another study by the same group, CXCL1 concentrations in CSF, but not in serum, were significantly higher in patients with meningoencephalitis than in those with meningitis (*p* < 0.01) early in the course of TBEV CNS involvement, and was the highest in a patient with the most severe clinical symptoms and altered consciousness. A similar (non-significant) trend was observed for IL-22, a mediator associated with Th17 responses, but not for IL-17A, IL-17F, the other Th17 mediators included in the study [34]. In our study which encompassed CSF and serum specimens obtained within the initial 14 days after the onset of meningoencephalitic phase of illness, many innate and Th1 adaptive cytokines and chemokines, particularly in CSF, correlated positively with the severity of acute illness including: innate mediators IFNα, IL-1β, TNF and CCL3 and Th1 mediators IL-12P40 and IL-12P70. In contrast, the levels of B cell chemoattractant CXCL12 in CSF correlated negatively with disease severity, whereas there were no correlations between any Th17 mediators in serum or CSF and TBE severity (Table 4). A possible explanation for this distinction between Th1 and Th17 immune responses and severity of acute disease is that at this early time point, the innate and Th1 responses were highly concentrated in CSF, the site of disease where immune responses could best exert their effect on symptomology, whereas Th17 responses and B cell responses tended to be concentrated in systemic circulation, outside the blood–brain barrier, where their effect on CNS-associated disease would be less pronounced. Additional support for this idea comes from correlations with leukocyte counts in CSF which showed that several innate and Th1 mediators in CSF correlated well with specific leukocyte counts in CSF, whereas Th17 mediators did not.

Our study has several limitations. First, we did not include controls. As CSF samples from healthy controls are difficult to obtain, this would limit the comparison of patients and controls to findings in serum. Moreover, we postulated that since TBE virus infects the CNS and causes neurological abnormalities, the immune responses to the virus, which are an important factor in the clinical presentation of disease, are likely concentrated in the CSF. Finally, the aim of this study was to better characterize the immune responses during TBE infection and assess if such responses may affect the clinical presentation of TBE. Thus, we compared the immune responses in CSF and serum in the same individual patients and we correlated these responses according to disease severity. We believe that the comparison of patients with the same infection, but different clinical presentation, was more appropriate than comparison with healthy controls or patients with other CNS infections. Second, because CSF was not centrifugated before freezing, chemokines/cytokines found in the CSF may partly account for intracellular cytokines released upon freeze-thawing of the samples. Third, since we performed sampling during the meningoencephalitic phase of TBE but did not obtain material from the illness prior to CNS involvement (Figure 1) we cannot exclude the possibility that differences between the immune responses in CSF and serum reflect the temporal lag between the events in the two compartments, i.e., that immune responses similar to those seen in CSF during the first 14 days of CNS involvement had been present in serum in the initial (pre-CNS) phase of the disease. However, it is difficult to capture such patients since most do not seek medical care or are not recognised as having the initial phase of TBE and are thus not diagnosed with TBEV infection until the meningoencephalitic phase of the disease. Furthermore, cell studies were based on standard leukocyte counts and a more detailed analysis of specific cell subsets is needed to characterize in greater detail the specific immune cell types involved and their activation status. Nevertheless, the design of our study enabled a comprehensive insight into the innate and acquired (T and B cells) immune responses in a large group of well-defined patients with TBE, and the assessment of the associations of the immune responses with the clinical course of TBE.

In conclusion, our findings demonstrate that during the meningoencephalitic phase of TBE, innate and Th1 adaptive inflammatory mediators and associated leukocytes are highly concentrated in CSF—the site of the disease. The consequence of this robust immune response is more severe acute illness, presumably by driving inflammation-induced tissue pathology. In contrast, inflammatory mediators associated with B cell responses, and particularly Th17 responses, appear concentrated in serum. These findings provide new insights in immunopathogenesis of TBE and implicate innate and Th1 adaptive responses in clinical presentation and severity of acute illness. Studies of specific cell subsets are needed to further elucidate the underlying immune mechanisms in TBE pathogenesis.

## Figures and Tables

**Figure 1 jcm-08-00731-f001:**
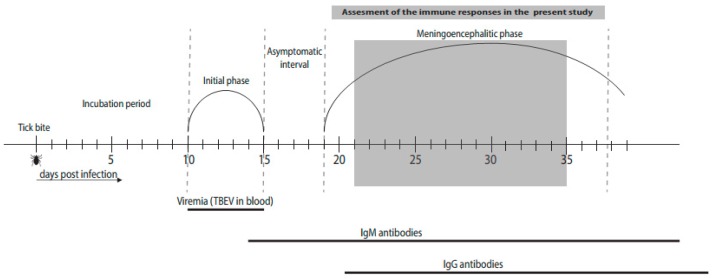
Schematic Diagram of the Course of Tick-borne Encephalitis. TBEV, tick-borne encephalitis virus.

**Figure 2 jcm-08-00731-f002:**
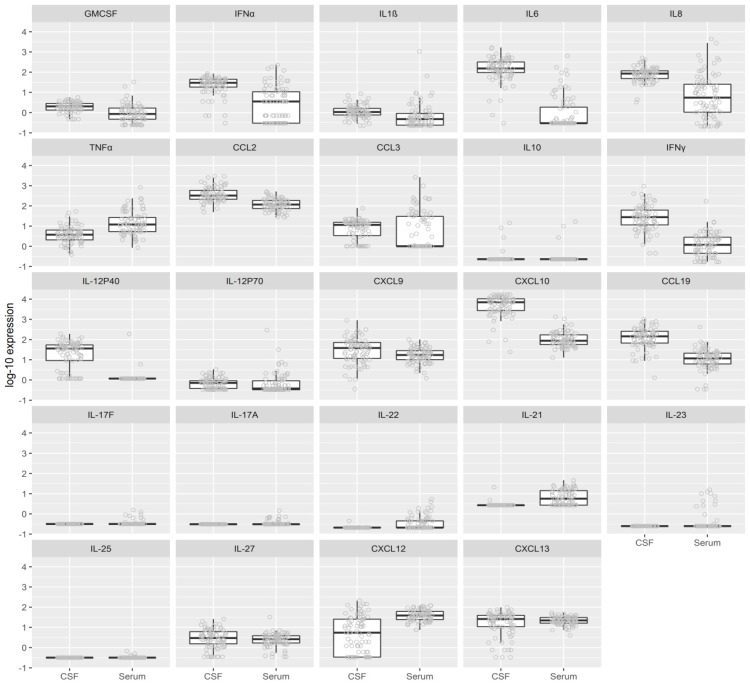
Levels of 24 Cytokines and Chemokines Associated with Innate or Adaptive Immune Responses Assessed in Matched Serum and Cerebrospinal Fluid (CSF) Samples Obtained on the Same Day during Meningoencephalitic Phase of Illness.

**Figure 3 jcm-08-00731-f003:**
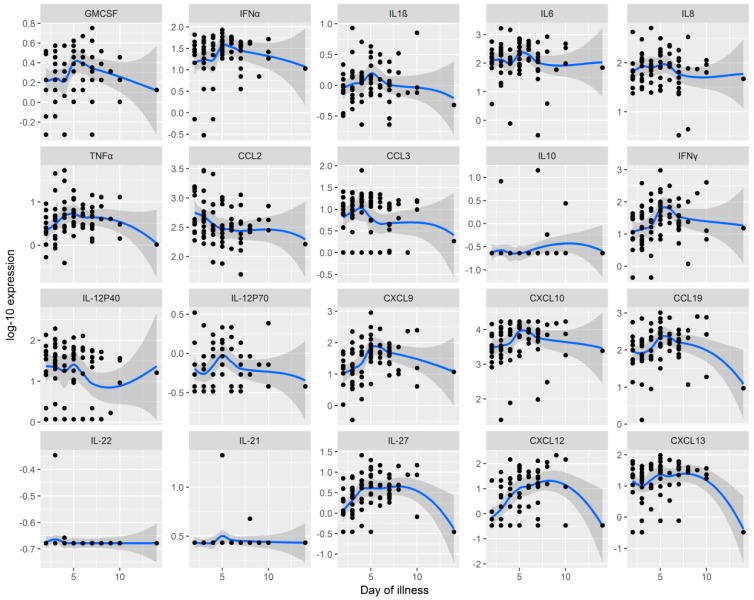
Dynamics Interplay between Inflammatory Immune Response in CSF and the Duration of the Early Meningoencephalitic Phase of the Illness. Four Representatives of Th17 Immune Response (IL-17, IL-17A, IL-23, and IL-25) are not Depicted Due to the Lack of Variability.

**Table 1 jcm-08-00731-t001:** Demographic, Clinical, and Laboratory Data on the Acute Illness for 81 Adult Patients with Tick-Borne Encephalitis in whom Levels of Cytokines and Chemokines in Serum and Cerebrospinal Fluid were Determined.

Characteristic	Number (%, 95% CI) or Median (IQR)
**Male sex**	40 (49.4; 38.1–60.7)
**Age (years)**	56 (43–63)
**Males**	58 (43–62.5)
**Females**	55 (45–63)
**Underlying illnesses**	39 (48.1; 36.9–59.5)
**Monophasic course of illness**	35 (43.2; 32.2–54.7)
**Clinical presentation**	
Meningitis	33 (40.7; 30.0–52.2)
Meningoencephalitis	40 (49.4; 38.1–60.7)
Meningoencephalomyelitis	8 (9.9; 4.4–18.5)
**Severity of illness**	
According to clinical assessment	
-Mild (meningitis)	33 (40.7; 30.0–52.2)
-Severe (meningoencephalitis or meningoencephalomyelitis)	48 (59.3; 47.8–70.1)
According to severity score	12 (4–23)
**Treatment in intensive care unit**	7 (8.6; 3.6–17.0)
**Duration (days)**	5 (3.5–9)
**Artificial ventilation: number; duration (days)**	2 (28.6; 3.7–71.0); 6 (5–7)
**Duration of illness before CSF and blood sample obtained (days) ^a^**	5 (3–6) ^b^
**Blood leukocyte count (× 10^9^ cells/L)**	9.9 (8.2–12.4) ^c^
Neutrophils (× 10^9^ cells/L)	7.7 (6.0–9.8)
Lymphocytes (× 10^9^ cells/L)	1.4 (1.1–1.8)
Monocytes (× 10^9^ cells/L)	0.7 (0.5–0.9)
**CSF leukocyte count (× 10^6^ cells/L)**	76 (37–134)
Neutrophils (× 10^6^ cells/L)	23 (11–48)
Lymphocytes (× 10^6^ cells/L)	43 (20–76)
Monocytes (× 10^6^ cells/L)	2 (0–5)
**CSF protein concentration (g/L)**	0.70 (0.53–0.91)
Elevated (> 0.45 g/L)	67 (82.7; 72.7–90.2)
**CSF glucose concentration (mmol/L)**	2.9 (2.6–3.3) ^c^
CSFglu/Sglu < 0.33	0/79 (0; 0–4.6)
**Albumin quotient (** **× 10^−3^) ^d^**	10.56 (7.98–12.92) ^b^
**IgG quotient (** **× 10^−3^) ^d^**	5.18 (3.98–6.62) ^b^
**Concomitant Lyme neuroborreliosis ^e^**	3/78 (3.8; 0.8–10.8)
**Positive *B. burgdorferi* sensu lato IgG antibodies ^f^**	7/78 (9.0; 3.7–17.6)

^a^ In patients with a biphasic course of illness, this figure was based on the time period from the beginning of the second (meningoencephalitic) phase until hospitalization; blood and CSF specimens were obtained on the same day within time span of up to a few hours; blood specimens for cytokine/chemokine determination and for leukocyte counts were obtained at the same blood collection. ^b^ Data available for 77 patients. ^c^ Data available for 80 patients. ^d^ Albumin (IgG) quotient between CSF and serum albumin (IgG) concentrations; albumin quotient was interpreted to be elevated when > 0.0074, IgG quotient when > 0.0035. ^e^ Demonstration of *B. burgdorferi* sensu lato infection of the central nervous system by isolation of *B. burgdorferi* sensu lato from CSF or intrathecal synthesis of *B. burgdorferi* sensu lato specific IgG or IgM antibodies. ^f^ The only marker of borrelial infection. CI, confidence interval; IQR, interquartile range; CSF, cerebrospinal fluid; CSFglu/Sglu, ratio of CSF and serum glucose concentrations.

**Table 2 jcm-08-00731-t002:** Cerebrospinal Fluid and Serum Concentrations of Cytokines/Chemokines Obtained in Patients During Meningoencephalic Phase of Tick-borne Encephalitis.

Cytokine/Chemokine	Concentrations (pg/mL) ^a^ Median (IQR)		*p*^b^ Value	*p*^c^ Value Adjusted
	CSF	Serum		
**Innate**				
GMCSF	**2 (1.3–2.9)**	0.87 (0.31–1.68)	**1.8 × 10^−9^**	**0.0001**
IFNα	**310 (18–45)**	0.71 (0.30–6.2)	**1.3 × 10^−11^**	**0.0001**
IL-1β	**1 (0.8–1.6)**	0.32 (0.23–0.69)	**4.2 × 10^−5^**	**0.0003**
IL-6	**154 (95–319)**	0.30 (0.30–0.76)	**9.0 × 10^−15^**	**0.0001**
IL-8	**85 (49–117)**	1.9 (0.62–15.1)	**5.3 × 10^−11^**	**0.0001**
TNFα	4 (2.1–6.4)	**6.4 (4.4–16.4)**	**3.0 × 10^−10^**	**0.0001**
CCL2	**325 (214–585)**	205 (158–259)	**6.1 × 10^−13^**	**0.0001**
CCL3	12 (3.4–15.4)	1.01 (1.0–8.8)	0.21	0.95
IL-10	0.2 (0.2–0.2)	0.2 (0.2 –0.2)	1	1
**Th1**				
IFNγ	**28 (11–62)**	1.1 (0.45–2.42)	**1.7 × 10^−14^**	**0.0001**
IL-12P40	**37 (9–54)**	1.2 (1.2–1.2)	**1.1 × 10^−12^**	**0.0001**
IL-12P70	0.7 (0.4–0.9)	0.38 (0.33–0.92)	0.0956	0.72
CXCL9	**39 (12–72)**	22 (12–41)	**0.0002**	**0.0013**
CXCL10	**7082 (2730–10451)**	91 (60–147)	**7.1 × 10^−15^**	**0.0001**
CCL19	**145 (67–259)**	24 (11–31)	**2.5 × 10^−13^**	**0.0001**
**Th17**				
IL-17F ^d^	320 (320–320)	**320 (320–320)**	**0.0059**	0.0711
IL-17A ^d^	0.31 (0.31–0.31)	**0.31 (0.31–0.31)**	**0.0011**	**0.0115**
IL-22	0.21 (0.21–0.21)	**0.22 (0.21–0.45)**	**3.8 × 10^−7^**	**0.0001**
IL-21	2.7 (2.7–2.7)	**5.71 (2.7–14.11)**	**4.7 × 10^−11^**	**0.0001**
IL-23 ^d^	250 (250–250)	**250 (250–250)**	**0.0017**	**0.0210**
IL-25 ^d^	320 (320–320)	320 (320–320)	0.0591	0.53
IL-27	3000 (1500–6100)	1500 (900–2100)	0.10	0.75
**B cell**				
CXCL12	5.5 (0.34–26)	**31 (19–53)**	**1.3 × 10^−9^**	**0.0001**
CXCL13	26 (11–39)	26 (19–36)	0.59	1

^a^ Final sample dilution for serum and CSF samples was the same (1:5). ^b^ The paired concentration levels of cytokines/chemokines in CSF and serum were compared using Wilcoxon signed rank test. ^c^ To control for false positives, the *p* values were adjusted using a multivariate permutation procedure [6]. ^d^ Medians and IQRs are the same but the other data (in the 1st and 4th quartiles) are not. Levels in compartment with significantly higher concentrations and the corresponding *p* values are shown in bold. IQR, interquartile range.

**Table 3 jcm-08-00731-t003:** Correlation of Cytokine and Chemokine Levels in Blood or Cerebrospinal Fluid (CSF) and White Cell Counts in Blood or CSF (Determined in the Meningoencephalic Phase of Tick-borne Encephalitis).

Immune Response Cytokine/Chemokine	Correlation with Leukocyte Count																	
	CSF Leucocytes									Serum Leucocytes								
	Neutrophils			Lymphocytes			Monocytes			Neutrophils			Lymphocytes			Monocytes		
	Rho ^a^	*p* Value ^b^	*p* Value ^c^	Rho ^a^	*p* Value ^b^	*p* Value ^c^	Rho ^a^	*p* Value ^b^	*p* Value ^c^	Rho ^a^	*p* Value ^b^	*p* Value ^c^	Rho ^a^	*p* Value ^b^	*p* Value ^c^	Rho ^a^	*p* Value ^b^	*p* Value ^c^
**Innate**																		
GMCSF	0.2511	0.02	0.88	0.2059	0.0652	0.99	0.1764	0.12	1	−0.1838	0.10	1	−0.0896	0.43	1	−0.1911	0.0896	0.99
IFNα	0.2003	0.0730	0.99	0.3575	0.0010	0.11	0.2576	0.0202	0.84	0.0040	0.97	1	−0.0060	0.96	1	0.0682	0.55	1
IL-1β	0.4768	<0.0001	**0.0008**	0.3007	0.0064	0.45	0.1801	0.11	1	0.0313	0.78	1	−0.0386	0.73	1	−0.0645	0.57	1
IL-6	0.5435	<0.0001	**0.0001**	0.1948	0.0814	0.99	0.1496	0.18	1	0.1231	0.28	1	−0.1520	0.18	1	−0.0391	0.73	1
IL-8	0.5858	<0.0001	**0.0001**	0.1346	0.23	1	0.0279	0.80	1	0.0481	0.67	1	0.0675	0.55	1	0.0218	0.85	1
TNFα	0.2669	0.0160	0.76	0.5344	<0.0001	**0.0001**	0.2009	0.0721	0.99	−0.0903	0.43	1	−0.0338	0.77	1	−0.0232	0.84	1
CCL2	0.3096	0.0049	0.37	−0.1752	0.12	1	0.0016	0.99	1	−0.2827	0.0110	0.63	−0.1539	0.17	1	−0.3189	0.0039	0.32
CCL3	0.1486	0.19	1	0.1244	0.27	1	0.2016	0.0711	0.99	−0.0532	0.64	1	−0.0740	0.51	1	0.0333	0.77	1
IL-10	0.1308	0.24	1	0.1025	0.36	1	−0.1588	0.16	1	−0.1134	0.32	1	−0.0579	0.61	1	−0.0989	0.38	1
**Th1**																		
IFNγ	0.2344	0.0352	0.96	0.5011	<0.0001	**0.0002**	0.2369	0.0333	0.95	−0.0671	0.55	1	−0.0518	0.65	1	−0.1119	0.32	1
IL-12P40	0.2011	0.0719	0.99	0.0345	0.76	1	0.2047	0.0668	0.99	−0.1502	0.18	1	−0.2486	0.0262	0.91	−0.1120	0.32	1
IL-12P70	0.2195	0.0490	0.99	0.3109	0.0047	0.36	0.2907	0.0085	0.54	0.0095	0.93	1	−0.1261	0.26	1	−0.0548	0.63	1
CXCL9	0.0573	0.61	1	0.4607	<0.0001	**0.0018**	0.2011	0.0719	0.99	−0.2458	0.0280	0.92	−0.2009	0.0739	0.99	0.0720	0.53	1
CXCL10	0.1951	0.0809	0.99	0.5968	<0.0001	**0.0001**	0.385	0.0004	**0.0422**	−0.2878	0.0096	0.59	−0.1808	0.11	1	−0.0943	0.41	1
CCL19	0.1533	0.17	1	0.4759	<0.0001	**0.0008**	0.3084	0.0051	0.38	−0.0325	0.77	1	0.0993	0.38	1	0.0163	0.89	1
**Th17**																		
IL-17F	NA	NA	NA	NA	NA	NA	NA	NA	NA	−0.0994	0.38	1	−0.0194	0.86	1	−0.1770	0.12	1
IL-17A	NA	NA	NA	NA	NA	NA	NA	NA	NA	0.0080	0.94	1	0.0802	0.48	1	−0.0079	0.94	1
IL-22	0.2278	0.0408	0.98	0.0228	0.84	1	−0.1850	0.0982	1	−0.0333	0.77	1	0.0484	0.67	1	0.0969	0.39	1
IL-21	−0.0937	0.41	1	0.13	0.25	1	−0.1850	0.0982	1	0.0154	0.89	1	−0.1430	0.21	1	0.0438	0.70	1
IL-23	NA	NA	NA	NA	NA	NA	NA	NA	NA	−0.0372	0.74	1	0.1385	0.22	1	−0.0618	0.59	1
IL-25	NA	NA	NA	NA	NA	NA	NA	NA	NA	−0.0365	0.75	1	−0.0726	0.52	1	-0.2147	0.0558	0.99
IL-27	0.2642	0.0172	0.79	0.4776	<0.0001	**0.0008**	0.1193	0.29	1	−0.0429	0.71	1	−0.2505	0.0250	0.90	−0.0811	0.47	1
**B cell**																		
CXCL12	−0.0106	0.93	1	0.5593	<0.0001	**0.0001**	0.1604	0.15	1	0.0936	0.41	1	0.1521	0.18	1	−0.1018	0.37	1
CXCL13	−0.1671	0.14	1	0.1607	0.15	1	0.1365	0.22	1	−0.1137	0.32	1	0.0316	0.78	1	0.0103	0.93	1

^a^ Spearman’s rho rank-based correlation; values > 0 indicate positive associations. ^b^ The association was tested using Spearman’s method. ^c^ Adjusted for multiple comparisons. Statistically significant *p* values are shown in bold. NA: not available (because no variability in concentrations was present).

**Table 4 jcm-08-00731-t004:** Correlation of Cytokine and Chemokine Levels in Serum or Cerebrospinal Fluid and the Severity of Tick-borne Encephalitis.

Immune Response Cytokine/Chemokine	Correlation with Severity * of Tick-Borne Encephalitis					
	CSF			Serum		
	Rho ^a^	Unadjusted *p* Value ^b^	Adjusted *p* Value ^c^	Rho ^a^	Unadjusted *p* Value ^b^	Adjusted *p* Value ^c^
**Innate**						
GMCSF	0.3402	0.0019	0.0570	0.3212	0.0035	0.0989
IFNα	0.2376	0.0327	0.62	0.2122	0.0572	0.82
IL-1β	0.3703	0.0007	**0.0207**	0.1461	0.19	0.99
IL-6	0.1368	0.22	0.99	0.1431	0.20	0.99
IL-8	0.1026	0.36	1	0.1159	0.30	1
TNFα	0.2507	0.0240	0.50	0.0518	0.65	1
CCL2	0.0049	0.97	1	0.215	0.0539	0.80
CCL3	0.363	0.0009	**0.0273**	0.0846	0.45	1
IL-10	0.082	0.47	1	−0.0834	0.46	1
**Th1**						
IFNγ	0.0572	0.61	1	0.1581	0.16	0.99
IL-12P40	0.3871	0.0004	**0.0128**	−0.009	0.94	1
IL-12P70	0.3591	0.0010	**0.0319**	0.2118	0.0577	0.83
CXCL9	0.1887	0.0916	0.94	0.4345	0.0001	**0.0022**
CXCL10	0.04	0.72	1	0.3516	0.0013	**0.0404**
CCL19	0.0316	0.78	1	0.0518	0.65	1
**Th17**						
IL-17F	NA	NA	NA	−0.0218	0.85	1
IL-17A	NA	NA	NA	−0.0112	0.92	1
IL-22	0.0664	0.56	1	−0.1627	0.15	0.99
IL-21	0.0087	0.94	1	−0.1155	0.30	1
IL-23	NA	NA	NA	0.0217	0.85	1
IL-25	NA	NA	NA	0.0177	0.88	1
IL-27	−0.0481	0.67	1	−0.0087	0.94	1
**B cell**						
CXCL12	−0.1087	0.33	1	−0.1067	0.34	1
CXCL13	−0.0879	0.44	1	−0.132	0.24	0.99

* Severity was determined quantitatively according to severity score. ^a^ Spearman’s rho rank-based correlation; values > 0 indicate positive associations. ^b^ The association was tested using Spearman’s method. ^c^ Adjusted for multiple comparisons using the permutation method of Westfall and Young [6]. Statistically significant *p* values are shown in bold. NA: not available (because no variability in concentrations was present).

**Table 5 jcm-08-00731-t005:** Correlation of Cytokine and Chemokine Levels and the Levels of IgG Antibodies against Tick-borne Encephalitis Virus in Cerebrospinal Fluid or Serum.

Immune Response Cytokine/Chemokine	Correlation with the Levels of Igg Antibodies against Tick-Borne Encephalitis Virus					
	CSF			Serum		
	Rho ^a^	Unadjusted *p* Value ^b^	Adjusted *p* Value ^c^	Rho ^a^	Unadjusted *p* Value ^b^	Adjusted *p* Value ^c^
**Innate**						
GMCSF	−0.13	0.25	1	−0.1914	0.0869	0.94
IFNα	0.0072	0.95	1	−0.1998	0.0736	0.91
IL-1β	−0.1316	0.24	1	−0.2310	0.0380	0.70
IL-6	−0.0627	0.58	1	−0.2476	0.0258	0.54
IL-8	−0.0253	0.82	1	−0.2409	0.0302	0.62
TNFα	0.1695	0.13	0.99	−0.2561	0.0210	0.47
CCL2	−0.2938	0.0078	0.22	−0.1659	0.14	0.99
CCL3	−0.3074	0.0052	0.15	−0.3493	0.0014	**0.0447**
IL-10	0.1548	0.17	0.99	0.0238	0.83	1
**Th1**						
IFNγ	0.2535	0.0224	0.49	0.0240	0.83	1
IL-12P40	−0.339	0.0020	0.0606	−0.0362	0.75	1
IL-12P70	−0.0756	0.50	1	−0.1534	0.17	0.9974
CXCL9	0.2650	0.0168	0.40	0.0093	0.93	1
CXCL10	0.2698	0.0149	0.36	−0.2431	0.0288	0.59
CCL19	0.2342	0.0353	0.68	0.2432	0.0287	0.59
**Th17**						
IL-17F	NA	NA	NA	−0.1104	0.33	1
IL-17A	NA	NA	NA	−0.1576	0.16	0.9957
IL-22	−0.1712	0.13	0.98	−0.0893	0.43	1
IL-21	0.1292	0.25	0.99	−0.1909	0.0878	0.94
IL-23	NA	NA	NA	−0.1310	0.24	1
IL-25	NA	NA	NA	−0.0364	0.75	1
IL-27	0.3563	0.0011	**0.0369**	−0.0941	0.40	1
**B cell**						
CXCL12	0.4462	0.0000	**0.0013**	−0.0805	0.47	1
CXCL13	0.1867	0.0952	0.95	−0.0528	0.64	1

* Severity was determined quantitatively according to severity score. ^a^ Spearman’s rho rank-based correlation; values > 0 indicate positive associations. ^b^ The association was tested using Spearman’s method. ^c^ Adjusted for multiple comparisons using the permutation method of Westfall and Young [6]. Statistically significant *p* values are shown in bold. NA: not available (because no variability in concentrations was present).

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
