# Peer review of "Inflammatory Immune Responses in the Pathogenesis of Tick-Borne Encephalitis"

_jcm, 2019, doi:10.3390/jcm8050731_

Round 1

Reviewer 1 Report

To date, the knowledge on pathogenesis of tick-borne encephalitis is somewhat limited. The article Inflammatory Immune Responses in the Pathogenesis of Tick-Borne Encephalitis by Bogovič and others characterized inflammatory immune responses in 81 Slovenian adult patients with TBE. The authors assessed the levels of 24 cytokines and chemokines associated with innate and adaptive T and B cell immune responses in matched serum and cerebrospinal fluid (CSF) samples. The inflammatory immune profiles implicated correlation in innate and Th1 adaptive responses in severity and clinical presentation of acute illness. The study material is very unique and well characterized previously. All in all the article is well written and the limitations of the study are discussed. However, there are some concerns, which are addressed below.

1.     Comparison of two different matrices (serum and CSF) is problematic methodologically, and TBE-specific changes cannot be interpreted. Please justify the comparison of serum and CSF concentrations without control group.

2.     Lines 73-74: The disease was monophasic in 35/81 of the cases, how were they defined as European TBEV subtype infections?

3.     Lines 121-124: How was the grouping performed? Not all are exclusively in designated groups (IL-10, CXCL12), which can distort interpretation of the results (e.g. B-cell association).

4.     Line 140: How was the association adjusted for multiple comparison?

5.     Lines 147-148: Were there differences in underlying diseases between females and males? Were there other infectious diseases?

6.     Table 1: Please clarify: “According to severity score”

7.     Line 174: Significantly higher in CSF except for TNF-alpha.

8.     Lines 174-175: “In contrast, mediators associated with Th17 and B cell immune responses were generally higher in serum.” Not clearly based on the table, except for IL-21 and CXCL12. Please correct.

9.     Line 183: Significantly higher concentrations are not shown in bold.

10.  Are Table 2 and Figure 2 presented from the same samples? What is the main difference in information presented?

11.  It is not clear to reader, which criteria was used to assess the correlation to disease severity presented in Table 4 (and Table 5?). Table 5 is identical to table 4. Clearer representation of the correlations needed. All the tables contain too much data for clarity (two P-values).

12.  Lines 173-174: Not all statistically significant. Please clarify.

Reviewer 2 Report

Design:

Healthy controls would be of interest. CSF may be difficult, however control serum would be nice to compare with. 

Text:

Text is well written. 

Figures and Tabes:

Tables are ok (can be improves and cleared up some). However, figures and legends are not clear. Plots are not easy to understand and would prefer another way to visualise differences. Values in plots need to be better described as well as legends. 

Author Response

Comments and Suggestions for Authors

Design:

Healthy controls would be of interest. CSF may be difficult, however control serum would be nice to compare with. 

Response: Since CSF samples from controls are difficult to obtain, this would limit the comparison of patients and controls to findings in serum. Moreover, we postulated that since TBE virus infects the CNS and causes neurological abnormalities, the immune responses to the virus, which are an important factor in the clinical presentation of disease, are likely concentrated in the CSF. Finally, the aim of this study was to better characterize the immune responses during TBE infection and assess if such responses may affect the clinical presentation of TBE. Thus, we compared the immune responses in CSF and serum in the same individual patients and we correlated these responses according to disease severity. We believe that the comparison of patients with the same infection but different clinical presentation was more appropriate than comparison with uninfected healthy controls. Information added. Please see lines 203-212.

Text:

Text is well written. 

Figures and Tabes:

Tables are ok (can be improves and cleared up some). However, figures and legends are not clear. Plots are not easy to understand and would prefer another way to visualise differences. Values in plots need to be better described as well as legends.

Response: We cleared up Table 2, Table 4, and Table 5.

 We added an explanation of the meaning of the figures in their legends and a paragraph describing the figures in the Methods section.

1.     In Methods

We displayed observed associations with outcome variables graphically by using box and whisker plots for categorical variables and scatter plots for numerical variables. We added a loess regression (locally weighted scatterplot smoothing) line with 95% CIs fitted by using the geom_smooth function in the ggplot2 R software. Please see lines 138 and 139, 141-145.

2.     In legend of Figure 2 we added

The gray points represent the individual data values measured in CSF and serum. The superimposed boxplots represent the first quartile (i.e., lower edge of the box), median (i.e., bar inside the box), third quartile (i.e., upper edge of the box), and minimum and maximum (i.e., length of the whiskers). If any points are at a greater distance from the quartiles than 1.5 times the interquartile range (IQR), the whisker length represent a distance of 1.5 times IQR from the upper or lower quartile. Please see lines 195-199.

3.     In the legend of Figure 3 we added

The black points represent individual values of the duration of Early Meningoencephalitic Phase of the Illness and the Cytokine and Chemokine Levels in CSF. The (loess) blue curves estimate the functional association between the two variables using a non-parametric smoother, the gray bands are their 95% CIs. Please see lines 18-20.

Reviewer 3 Report

Manuscript Manuscript ID: jcm-482882 “Inflammatory Immune Responses in the Pathogenesis of Tick-Borne Encephalitis”

The paper by Bogovic et al. is a study in which cytokines and chemokines are measured from 81 patients in matched cerebrospinal fluid and serum samples, together with an evaluation of clinical symptoms and outcome in the same patients. The authors used bead-based multiplex assays to evaluate 24 chemokines and cytokines, that they divided into groups of association with innate, adaptive Th1 and B cell immune response.

This study is the first comprehensive study on matched cerebrospinal fluid and serum samples on multiple cytokines/chemokines evaluated simultaneously during TBE. Also, the carefully described cohort makes it possible to correlate the lab data with clinical parameters. Thus, the study is interesting, however there are some concerns with the study that would need to be clarified from the authors.

1. What was the rationale for choosing the particular cytokines and chemokines for the study that were used this would need to be added into the manuscript and carefully discussed?

2. It is not clear how the authors made the division of the different cytokines and chemokines into sub-groups, as there is a lack of references for this, this would need to be added into the manuscript and carefully discussed.

3. In the material and methods section it is not stated whether the cerebrospinal fluid is cell free (was the sample centrifugated to get rid of the cells before freezing?), or may the chemokines/cytokines found in the CSF also account for intracellular cytokines that would be released upon freeze-thawing of the samples. This should be stated in the M&M section, and the influence of this should be discussed.

4. In order to know if the enhanced cytokines/chemokines classified as innate and Th1 adaptive responses is specific for TBE immunopathogenesis, controls such as patients with aseptic meningitis or borreliosis or healthy control should be used.

5. Results/Tables and Figures

Table 1. Are the numbers in the table correct for the albumin quotient correct (10E-3?)?

Table 2. Is the median and IQR correct for IFN-alpha in CSF? Also, are the values for IL-17F and IL-17A correct (you comment in the table that the other data are not the same?)? What is the limit of detection in this assay (low and high)?

Table 3. This table contains very much information and the content would probably be better illustrated in a different way, for example with correlation figures.

Table 4. This table contains very much information and the content would probably be better illustrated in a different way, for example with correlation figures. If the authors want to keep this table, the significant values should be marked in bold (as in table 3).

Table 5. Is this the same table as Table 4? Looks identical.

Are IL-27 and CXCL12 statistical significant as stated in the results section?

Figure 3. What statistics were used to determine that CXCL12 and IL-27 levels was associated with duration of illness?

6. In the discussion, the authors seem to mix the immune response (e.g cellular response) with the inflammatory response (e.g. cytokines/chemokines mediated by different cell types in the immune response). For example, on line 121 the authors correctly state that studies of cellular immune responses in human patients during the initial phase of TBE are lacking. In conjunction with this sentence the authors then writes that several articles have characterized the immune response to TBEV in the second phase of the disease and make reference to a number of studies studying cytokines and chemokines (inflammation) during TBE. The authors should try to correctly reference the current literature on the immune response during TBE or re-write the sentence on line 121.

7. The authors should focus the discussion more on their results, the meaning and implications of the findings.

Author Response

The responses to reviewer 3 are now provided in the word

Round 2

Reviewer 1 Report

All the questions are addressed in the revised manuscript and provide now a clearer presentation of the study.

Reviewer 2 Report

The authors have revised the manuscript according to comments. It is well written, and the data are described in a clear way. 

Reviewer 3 Report

Manuscript Manuscript ID: jcm-482882 “Inflammatory Immune Responses in the Pathogenesis of Tick-Borne Encephalitis”

Bogovic et al. have made some revisions to the manuscript, but have rather focused on responding to the reviewer’s questions than making revisions in the manuscript for the text and reference part. For example, for the reviewer (#3) question 1 and 2, only 2 references on Th17 (reference 36-37) were added to the text and the rationale for choosing and the division of the cytokines and chemokines were not at all introduced in the paper. For a paper that is built on analyzing differences of cytokines and chemokines associated with innate and Th1 adaptive immune responses one would expect that a background and appropriate references is given to the reader. 

In addition, the authors did not put a lot of effort into focusing their discussion, but rather only added three sections into the discussion. 

The correlation figures is now shown as S1-S6. One can wonder if correlation can be performed for the data points in S3. For example, within figure S3 with one outlier, is there really a positive correlation for the number of monocytes and CXCL10 concentration?